

# Criteria for assessing the adaptive changes in mixed martial arts (MMA) athletes of strike fighting style in different training load regimes

Andrii Chernozub[1], Veaceslav Manolachi[2,3], Georgiy Korobeynikov[4], Vladimir Potop[3,5], Liudmyla Sherstiuk[1], Victor Manolachi[2,3] and Ion Mihaila[5]

[1] Petro Mohyla Black Sea National University, Mykolaiv, Ukraine
[2] "Dunarea de Jos" University of Galati, Galati, Romania
[3] State University of Physical Education and Sport, Chisinau, Republic of Moldova
[4] National University of Physical Education and Sport, Kyiv, Ukraine
[5] Department of Physical Education and Sport, University of Pitesti, Pitesti, Romania

Corresponding author
Veaceslav Manolachi,
manolachivsciences@yahoo.com

## ABSTRACT

**Background**. To study the peculiarities of changes in functional indicators and body composition parameters of mixed martial arts (MMA) athletes of strike fighting style and a number of biochemical blood indicators during two months of using different intensity training load regimes, and to determine the most informative criteria for assessing adaptive body changes in these training conditions.

**Methods**. We examined 40 MMA athletes (men) aged 20–22, who used mainly strike fighting style in their competitive activity, and divided them into 2 research groups (A and B), 20 athletes in each group. Group A athletes used medium intensity ($R_a = 0.64$), and group B—high intensity ($R_a = 0.72$) training load regime. To assess the adaptive body changes we applied methods of control testing of maximum muscle strength growth (1 RM), special training (the number of accurate kicks on the mannequin for 30 s), and bioimpedansometry. By monitoring biochemical parameters (testosterone, cortisol, creatinine, phosphorus, calcium, cholesterol, LDH) in the blood serum of athletes, we determined the peculiarities of adaptive-compensatory body reactions in response to training loads.

**Results**. The obtained results of special training increased during the study period by an average of 10.5% in group B athletes, but group A participants' results had no significant changes compared to basal data. The largest increase in the development of maximum muscle strength growth by an average of 44.4% was recorded after 2 months of research in group B. Group B athletes also had positive changes in body fat and fat-free mass indicators during the study which were two times higher than the results of group A. The laboratory studies and correlation analysis showed informative biochemical markers (cortisol, testosterone and creatinine) for assessing the condition of athletes in both groups before using high and medium training load regimes. The biochemical markers for assessing the adaptive-compensatory reactions of athletes in response to high-intensity physical stimuli at the beginning of the study were indicators of LDH and cholesterol, and in conditions of medium intensity it was LDH, testosterone and cortisol. After 2 months of study the set of biochemical markers assessing the adaptation processes before the load completely changed only in group B athletes and consisted of

LDH, phosphorus, cholesterol, and calcium. At the same time, the set of biochemical criteria for assessing adaptive-compensatory reactions after training in group B athletes was completely changed compared with the data recorded at the beginning of the study. **Conclusion**. Defining the optimal set of criteria for assessing the adaptive-compensatory changes in MMA athletes of strike fighting style will allow in the shortest possible time to correct the parameters of the training load regime for accelerating the body functionality in the process of special power training.

## INTRODUCTION

Leading experts in sports and physical training (*James et al., 2016a*; *James et al., 2016b*; *Kirk et al., 2021*; *Tota & Wiecha, 2022*) have recently showed great interest in optimizing the system of integrated control of athletes' adaptation processes in mixed martial arts (MMA) to increase the body functional reserves, to improve the mechanisms of training activity correction, and to maximize performance in the shortest possible time. The complexity of the process of developing an optimal set of criteria for assessing adaptive body changes of MMA athletes is connected with taking into account their individual characteristics, level of training, the style of fighting, and parameters of training load regimes used during training (*Matthews & Nicholas, 2017*; *Slimani et al., 2017*; *Chernozub et al., 2019*).

The wide and diverse range of techniques, methods, and principles of training activities used by athletes, depending on the style of fighting in mixed martial arts, raises a number of controversial issues among experts on determining effective training load parameters and correction mechanisms (*Del Vecchio & Franchini, 2011*; *Chernozub et al., 2019*; *Kirk et al., 2021*). The search for new ways to improve the training of athletes of strike fighting style requires the development of explosive strength and endurance, and a constant increase in adaptive reserves due to intra-muscular and intermuscular coordination and energy efficiency of muscle activity in anaerobic training load regime (*James et al., 2016a*; *James et al., 2016b*; *Futorniy et al., 2016*).

Several researchers (*Manolachi, 2015*; *Korobeynikov et al., 2017*; *Titova et al., 2018*; *Chernozub et al., 2020*) considering speed and power training of athletes to be the key factor of adaptive changes effectiveness, noted that the main criteria for assessing the effectiveness of different intensity training load on the body's functional reserves and the level of athletes' fitness are the following indicators: morpho-functional parameters, assessment of special training and changes in biochemical components of blood. At the same time, the problem of searching the effective training load regimes in the process of special strength training of MMA athletes of strike fighting style has not been studied yet and the ways of determining the optimal set of informative criteria for assessing adaptive body changes for further correction of the training process have not been described in scientific literature.

Extensive use of biochemical control indicators (hormones, enzymes, microelements, *etc.*) to assess the adaptive-compensatory reaction of athletes in training and competitive activities is an integral part of the training system in martial arts and fitness (*Marques et al., 2017*; *Philippou et al., 2017*; *Wochyński & Sobiech, 2017*; *Titova et al., 2018*; *Kılıç et al., 2019*; *Tota & Wiecha, 2022*) in recent decades. At the same time there is no clear definition which indicators of biochemical blood control reliably reflect the course of adaptive changes depending on the structure of special strength training and the peculiarities of the training load regime for MMA athletes of strike fighting style.

One of the priority issues of improving the training system in MMA is the development of mechanisms for operational control of the functional state of athletes using the minimum number of indicators but the most informative set of biochemical blood markers. Appropriate ways to optimize the diagnosis of athletes will reduce the time to determine the necessary mechanisms for correcting the value of the training load regime and decrease the cost of research.

The purpose of the study is to review the peculiarities of changes in functional indicators and body composition parameters of MMA athletes of strike fighting style and a number of biochemical blood indicators during two months of using different intensity training load regimes, and to determine the most informative criteria for assessing adaptive body changes in these training conditions.

## MATERIALS & METHODS

### Participants

We examined 40 MMA athletes (men) aged 20–22 with an average training experience is $2.5 \pm 0.32$ years. The research took place at the stage of specialized basic training of athletes. These athletes were divided them into two research groups (A and B), 20 athletes in each group. Thus, group A athletes used medium intensity training load regime ($R_a = 0.64$), and group B representatives used high intensity training load regime ($R_a = 0.72$).

For the validity of study, we selected participants without significant differences in the initial parameters of the studied indicators.

The experimental study was approved by Petro Mohyla Black Sea National University Ethics Committee for Biomedical Research in accordance with the Ethical Standards of the Helsinki Declaration (ecbr10-02-2022). The participants gave written consent to the study in accordance with the recommendations of the Biomedical Research Ethics Committees (*World Health Organization, 2000*).

We used diagnostic equipment of the medical center of the university for medical examination, assessment of morpho-functional parameters, and biochemical control of blood serum in athletes during the study.

### Measurements
#### Maximal muscle strength

Control testing of the maximum muscle strength growth (1 RM) in athletes of both groups took place at the beginning and after 2 months of special strength training using the specified training load regimes. Control testing consisted of the following control

exercises: bench press on the Smith simulator, block thrust behind the head, legs press in the simulator lying. Measurements of 1 RM were performed during control exercises on simulators to reduce the level of injuries of study participants. During the control testing and throughout the training period, strength exercises were performed in accordance with generally accepted techniques and methods (*James et al., 2016a*; *James et al., 2016b*; *Loturco et al., 2021*).

## Special training

The special training development of group participants was determined using the method of control testing of the training level in MMA athletes of the strike fighting style. The proposed method is based on determining the maximum number of kicks on the mannequin during the control exercises (front kick, side kick, reverse side kick, roundhouse kick) for 30 s following the appropriate technique. We chose such control exercises because of their frequent usage in the competitive activity of mixed martial arts.

## Body composition

A non-invasive biophysical method of bioimpedansometry was used to determine the body composition of athletes during the study (initial data, after the first and second months of training). This method is based on measuring the electrical resistance of biological tissues of the body. Determination of the body composition parameters occurs in the process of computer processing of the obtained data. The method of bioimpedansometry allows to determine the following indicators: body fat (BF, %), fat-free mass (FFM, kg), body cell mass (BCM, %), body mass index (BMI, cond. units), dry cell weight (DCW, kg). To evaluate the studied parameters, we used a bioimpedance analyzer (body composition analyzer) consisting of diagnostic computerized hardware-software complex KM-AP-01 (Diamond-AST configuration) (VYUSK. 941118.001 PE) (*Martyrosov, Nikolayev & Rudnev, 2006*).

## Biochemical parameters

The activity of lactate dehydrogenase (LDH) and the concentration of creatinine in the blood serum of athletes were determined by the kinetic method on the equipment High Technology, Inc. (North Attleboro, MA, USA) with a set of reagents (PRESTIGE 24i LQ LDH) (Poland). Testosterone and cortisol concentration in the blood serum was determined by enzyme-linked immunosorbent analysis, using a set of reagents (SteroidIFA-testosterone) on the equipment of Alcor Bio Ltd. (St. Petersburg, Russia). The concentration of calcium and phosphorus in the blood was determined using a photometric method by measuring the optical density on a spectrophotometer StatFax 4700 using a set of reagents (Medical Analysis) for determining calcium, and a set of reagents Liquick Cor-PHOSPHORUS (Poland) for determining the concentration of phosphorus. The concentration of total cholesterol was determined by colorimetric and enzymatic methods with esterase and cholesterol oxidase using Liquick Cor-CNOL diagnostic kit (Liquick, Warszawa, Poland). The blood sampling was performed according to the general requirements of biomedical research (*Tietz, Finley & Pruden, 1995*). Venous

**Table 1  Structure of training load regimes used by participants of research in the process of strength training in MMA.**

| Training load indicators | Training load regimes | |
| --- | --- | --- |
| | Medium intensity ($R_a = 0.64$) | High intensity ($R_a = 0.72$) |
| Amplitude of exercise, %: | Maximum (100%) | Partial (90% from maximum) |
| Conditional coefficient of movement amplitude (Q) | 0.9 | 0.8 |
| Duration of movement phases | Concentric phase–2 s; | Concentric phase–3 s; |
| | Eccentric phase–3 s | Eccentric phase–6 s |
| Duration of one repetition (t, s) | 5 | 9 |
| Duration of rest between sets, p | 60 | 40 |
| Number of repetitions in a set | 8 | 4 |
| Maximum duration of work in a set ($T_{max}$, s) | 40 | 36 |
| Projectile working mass (m), % of maximum (1RM) | 64–69 | 72–74 |
| Training load index ($R_a$) | 0.64 | 0.72 |
| Training load intensity | Medium intensity | High intensity |
| Duration of training session, min | 30–32 | 30–32 |

blood sampling was performed by a paramedic before and after training at the beginning and end of the second month of the study in compliance with all requirements and norms.

## Experimental design

The research took place in several stages:

In the first stage we determined the characteristics of adaptive-compensatory reactions in MMA athletes of strike fighting style who used different intensity of training load regimes (Table 1), which were effective in power fitness (*e.g.*, *Chernozub et al., 2020*), promoted accelerated growth of maximum muscle strength, and stimulated pronounced adaptive body changes. Thus, group A participants used medium intensity training load regime ($R_a = 0.64$), and group B athletes used high intensity training load regime ($R_a = 0.72$) for 2 months of research with a frequency of three sessions per week. During each workout, no more than 3–4 muscle groups were loaded with two exercises (basic and isolated) for each. Training exercises were performed on simulators following the techniques that we defined in order to prevent injuries.

In the second stage, we studied the nature of adaptive body changes to the stress stimulus (power loads in a training regime), we also determined peculiarities of changes in maximum muscle strength and the level of special training. We studied changes in the concentration of cortisol, testosterone, phosphorus, calcium, cholesterol, creatinine, and lactate dehydrogenase activity in the blood serum of both group athletes in conditions of different intensity training load regimes. We compared the results of the studied indicators to determine the optimal training load regime for MMA athletes of strike fighting style, which could maximally promote the processes of adaptation in the shortest possible time.

In the third stage we compared the results of the correlation between the biochemical parameters of blood serum before and after training in the given conditions of muscular activity at the beginning and end of 2-month study. The obtained results allowed to determine which of the studied biochemical indicators of blood serum had multiple strong

**Table 2 Parameters of maximum muscle strength (1 RM) development in both group participants during the research, $n = 40$.**

| Control exercises, 1RM, kg | Group | Observation period, months | | | $\chi^2, p$ $df = 2$ |
| --- | --- | --- | --- | --- | --- |
| | | Initial data | After 1 month of training | After 2 months of training | |
| Bench press on the Smith simulator | A | $62.47 \pm 1.72$ | $73.62 \pm 1.17^*$ $Z=-3.8; p < 0.000$ | $85.00 \pm 2.21^*$ $Z=-3.9; p < 0.000$ | $\chi^2=39.5$ $p < 0.000$ |
| | B | $61.50 \pm 1.74$ | $74.62 \pm 1.89^*$ $Z=-3.9; p < 0.000$ | $85.75 \pm 1.89^*$ $Z=-3.9; p < 0.000$ | $\chi^2=40.0$ $p < 0.000$ |
| Block thrust behind the head | A | $65.25 \pm 1.39$ | $72.50 \pm 1.02^*$ $Z=-3.7; p < 0.000$ | $78.12 \pm 1.21^*$ $Z=-4.1; p < 0.000$ | $\chi^2=39.1$ $p < 0.000$ |
| | B | $60.00 \pm 1.52$ | $69.37 \pm 1.51^*$ $Z=-3.7; p < 0.000$ | $77.75 \pm 1.38^*$ $Z=-3.8; p < 0.000$ | $\chi^2=38.5$ $p < 0.000$ |
| Legs press in the simulator lying | A | $115.25 \pm 3.68$ | $143.52 \pm 5.01^*$ $Z=-3.9; p<0,000$ | $169.62 \pm 6.27^*$ $Z=-3.9; p < 0.000$ | $\chi^2=40.0$ $p < 0.000$ |
| | B | $122.20 \pm 4.58$ | $167.00 \pm 5.21^*$ $Z=-3.9; p < 0.000$ | $201.25 \pm 4.98^*$ $Z=-3.9; p < 0.000$ | $\chi^2=40.0$ $p < 0.000$ |

**Notes.**
*The difference in comparison with previous results is significant according to the Wilcoxon test ($p < 0.05$); df, the number of degrees of freedom; $p$, level of significance.

correlations at different phases of control in each group. This analysis revealed the optimal set of informative criteria for assessing short-term and long-term adaptation for MMA athletes using strike fighting style.

## Statistical analysis

Statistical analysis of the study results was performed using the software package IBM $^\star$ SPSS $^\star$ Statistics 23 (StatSoft, Inc., Tulsa, OK, USA). Descriptive statistics methods were used to calculate the arithmetic mean and the error of the mean. The non-parametric Wilcoxon test was used to assess the reliability of paired differences, and Friedman's ANOVA was used to analyze repeated measurements. The relationship between certain variables and individual-typological characteristics of the subjects was established using Spearman's rank correlation coefficients (*e.g.*, *Nasledov, 2013*).

## RESULTS

Table 2 presents the results of the dynamics of maximum muscle strength development (1 RM) in both group participants during 2 months of research using different intensity training load regimes.

The results analysis showed that both group athletes demonstrated positive dynamics of maximum muscle strength growth performing the control exercises during the research. Thus, using high intensity training load regime ($R_a = 0.72$) by group B athletes increased the studied indicator fixed in control exercises by 44.4% ($p < 0.05$) on average for 2 months of training compared to initial data. Group A athletes who used medium intensity training load regime ($R_a = 0.64$) also showed positive changes of the investigated indicator (1 RM) for the similar period of time which made up 34.3% ($p < 0.05$).

**Table 3** Parameters of the number of accurate kicks on the mannequin for 30 s in both group participants during the research, $n = 40$.

| Control exercises | Group | Observation period, months | | | $\chi^2, p$ $df = 2$ |
|---|---|---|---|---|---|
| | | Initial data | After 1 month of training | After 2 months of training | |
| Front kick | A | $14.25 \pm 0.23$ | $14.40 \pm 0.21$ $Z=-1.0; p > 0.05$ | $14.45 \pm 0.19$ $Z=-0.4; p > 0.05$ | $\chi^2=2.3$ $p > 0.05$ |
| | B | $14.10 \pm 0.17$ | $14.65 \pm 0.19^*$ $Z=-2.4; p < 0.05$ | $15.75 \pm 0.27^*$ $Z=-3.3; p < 0.001$ | $\chi^2=22.9$ $p < 0.000$ |
| Side kick | A | $18.60 \pm 0.19$ | $18.90 \pm 0.19$ $Z=-1.5; p > 0.05$ | $18.95 \pm 0.18$ $Z=-0.4; p > 0.05$ | $\chi^2=5.3$ $p > 0.05$ |
| | B | $18.45 \pm 0.16$ | $19.05 \pm 0.18^*$ $Z=-2.1; p < 0.05$ | $19.80 \pm 0.25^*$ $Z=-2.8; p < 0.005$ | $\chi^2=15.1$ $p < 0.001$ |
| Reverse side kick | A | $16.40 \pm 0.18$ | $16.75 \pm 0.14$ $Z=-1.8; p > 0.05$ | $16.95 \pm 0.16$ $Z=-1.6; p > 0.05$ | $\chi^2=9.4$ $p<0.009$ |
| | B | $16.50 \pm 0.13$ | $17.25 \pm 0.17^*$ $Z=-2.9; p < 0.005$ | $18.45 \pm 0.11^*$ $Z=-3.2; p < 0.001$ | $\chi^2=27.3$ $p < 0.000$ |
| Roundhouse kick | A | $18.05 \pm 0.16$ | $18.40 \pm 0.11$ $Z=-1.8; p > 0.05$ | $18.50 \pm 0.11$ $Z=-0.8; p > 0.05$ | $\chi^2=6.4$ $p<0.04$ |
| | B | $18.15 \pm 0.15$ | $18.60 \pm 0.13^*$ $Z=-2.2; p < 0.05$ | $20.20 \pm 0.15^*$ $Z=-3.9; p < 0.000$ | $\chi^2=31.8$ $p < 0.000$ |

**Notes.**
*The difference in comparison with previous results is significant according to the Wilcoxon test ($p < 0.05$); df, the number of degrees of freedom; p, level of significance.

The peculiarities of the change in the indicators of special training in Mixed Martial Arts (the number of accurate kicks on the mannequin for 30 s) for 2 months of research are presented in Table 3.

We observed that both group representatives had no significant differences in control indicators at the beginning of research, which allowed to determine the degree of impact of the proposed training load regimes on their level of special training.

The parameters of special training (the number of accurate kicks on the mannequin for 30 s) during control exercises increased by an average of 10.5% ($p < 0.05$) during the study period in group B athletes, who used high intensity training load regime ($R_a = 0.72$), compared to baseline data. At the same time, the studied indicators in group A athletes showed only a positive unreliable tendency to change by an average of 2.2% ($p > 0.05$) for the same period of training in conditions of medium intensity training load regime ($R_a = 0.64$).

Table 4 shows the peculiarities of changes in body composition of athletes of both groups in the given conditions of muscular activity during 2 months of research.

The bioimpedansometry results analysis showed that the body fat parameter positively decreased by 13.5% ($p < 0.05$) for 2 months in group B athletes. Group A representatives also decreased the studied indicator, but it was by almost 4 times less than group B.

During the study, the fat-free mass parameter increased by 3.2% ($p < 0.05$) in athletes of group B compared with baseline data. Representatives of group A showed no significant changes in the studied indicator.

**Table 4 Parameters of body composition of research participants during two months, $n = 40$.**

| Parameters | Group | Observation period, months | | | $\chi^2, p$ $df = 2$ |
|---|---|---|---|---|---|
| | | Initial data | After 1 month of training | After 2 months of training | |
| BF (body fat, %) | A | 11.93 ± 0.83 | 13.25 ± 0.84* Z=−2.5; $p < 0.01$ | 11.47 ± 0.80* Z=−2.3; $p < 0.05$ | $\chi^2$=6.1 $p < 0.05$ |
| | B | 16.30 ± 1.08 | 14.27 ± 1.18* Z=−3.4; $p < 0.001$ | 14.09 ± 1.06 Z=−1.1; $p > 0.05$ | $\chi^2$=13.6 $p < 0.001$ |
| FFM (fat-free mass, kg) | A | 67.50 ± 1.29 | 67.05 ± 1.28 Z=−1.3; $p > 0.05$ | 68.62 ± 1.25 Z=−1.6; $p > 0.05$ | $\chi^2$=2.1 $p > 0.05$ |
| | B | 63.86 ± 0.62 | 65.77 ± 0.94* Z=−3.7; $p < 0.000$ | 65.94 ± 0.99 Z=−0.5; $p > 0.05$ | $\chi^2$=28.2 $p < 0.000$ |
| BCM (body cell mass, %) | A | 62.54 ± 0.71 | 65.31 ± 1.11* Z=−2.8; $p < 0.005$ | 63.70 ± 0.24 Z=−1.3; $p > 0.05$ | $\chi^2$=6.2 $p < 0.05$ |
| | B | 63.98 ± 0.16 | 62.45 ± 0.31* Z=−3.5; $p < 0.000$ | 62.84 ± 0.22 Z=−0.4; $p > 0.05$ | $\chi^2$=12.9 p<0.005 |
| BMI (body mass index, cond. units) | A | 23.80 ± 0.51 | 23.76 ± 0.39 Z=−1.4; $p > 0.05$ | 23.92 ± 0.46 Z=−1.4; $p > 0.05$ | $\chi^2$=6.8 $p < 0.05$ |
| | B | 24.09 ± 0.58 | 24.27 ± 0.60* Z=−2.0; $p < 0.05$ | 24.39 ± 0.58 Z=−0.4; $p > 0.05$ | $\chi^2$=6.1 $p < 0.05$ |
| DCW (dry cell weight, kg) | A | 10.95 ± 0.29 | 11.35 ± 0.25* Z=−2.5; $p < 0.05$ | 11.34 ± 0.22 Z=−0.7; $p > 0.05$ | $\chi^2$=13.4 $p < 0.001$ |
| | B | 10.55 ± 0.11 | 10.50 ± 0.14* Z=−3,2; $p < 0.001$ | 10.69 ± 0.13* Z=−3.9; $p < 0.000$ | $\chi^2$=38.1 $p < 0.000$ |

**Notes.**

*The difference in comparison with previous results is significant according to the Wilcoxon test ($p < 0.05$); df, the number of degrees of freedom; $p$, level of significance.

Table 5 presents the results of changes in biochemical inadicators of the blood serum in the participants of both groups in terms of muscular activity, taking into account the peculiarities of training load regimes during all stages of the research.

The concentration of cortisol in the blood serum of group A athletes in response to medium intensity training load regime ($R_a = 0.64$) remained almost unchanged at all stages of control. Group B athletes demonstrated increase in the concentration of this hormone by 57.9% ($p < 0.05$) at the beginning and 32.9% ($p < 0.05$) after 2 months of study in response to high intensity training load regime ($R_a = 0.72$).

The control results of testosterone concentration in the blood serum showed that the basal level of this hormone increased by 30.4% ($p < 0.05$) in athletes of group A, and by 20.0% ($p < 0.05$) in participants of group B.

Using high intensity training load regime ($R_a = 0.72$) for 2 months stimulated an increase of the basal level of creatinine in the blood serum by 17.4% ($p < 0.05$) in group B participants. At the same time the level of creatinine in group A participants remained unchanged.

Calcium concentration was observed only in athletes of group A in response to stress stimuli at the beginning of the study (by 5.2% compared to rest) and after 2 months (by 2.5%) while using medium intensity training load regime ($R_a = 0.64$). At the same time, the results of this biochemical indicator did not change significantly in group B representatives.

**Table 5  Biochemical blood indicators of group participants during the research, $n = 40$.**

| Indicator | Group | At the beginning of the study | | After 2 months of training | |
|---|---|---|---|---|---|
| | | Before training load | After training load | Before training load | After training load |
| Cortisol, nmol/l | A | $320.53 \pm 14.72$ | $351.62 \pm 27.85$ Z=−0.5; $p > 0.05$ | $416.50 \pm 26.70^{**}$ Z=−2.5; $p < 0.01$ | $506.85 \pm 42.06$ Z=−0.7; $p > 0.05$ |
| | B | $332.72 \pm 11.26$ | $525.55 \pm 36.41^{*}$ Z=−3.9; $p < 0.000$ | $509.88 \pm 28.79^{**}$ Z=−3.6; $p < 0.000$ | $677.72 \pm 9.50^{*}$ Z=−3.9; $p < 0.000$ |
| Testosterone, nmol/l | A | $19.42 \pm 1.28$ | $20.52 \pm 1.72$ Z=−1.0; $p > 0.05$ | $25.32 \pm 2.27^{**}$ Z=−2.9; $p < 0.005$ | $27.65 \pm 1.98^{*}$ Z=−3.3; $p < 0.001$ |
| | B | $28.85 \pm 1.53$ | $29.15 \pm 1.63^{*}$ Z=−2.1; $p < 0.05$ | $34.62 \pm 0.62^{**}$ Z=−3.3; $p < 0.001$ | $35.72 \pm 0.75$ Z=−1.8; $p > 0.05$ |
| Creatinine, μmol/l | A | $85.50 \pm 0.83$ | $83.95 \pm 0.86$ Z=−1.7; $p > 0.05$ | $86.00 \pm 1.27$ Z=−0.5; $p > 0.05$ | $94.85 \pm 1.25^{*}$ Z=−3.9; $p < 0.000$ |
| | B | $84,60 \pm 2.24$ | $82.15 \pm 2.88$ Z=−0.8; $p > 0.05$ | $99.30 \pm 1.49^{**}$ Z=−3.9; $p < 0.000$ | $105.75 \pm 1.16^{*}$ Z=−3.9; $p < 0.000$ |
| Calcium, mmol/l | A | $2.48 \pm 0.01$ | $2.61 \pm 0.11^{*}$ Z=−3.9; $p < 0.000$ | $2.36 \pm 0.01^{**}$ Z=−3.9; $p < 0.000$ | $2.42 \pm 0.01^{*}$ Z=−3.9; $p < 0.000$ |
| | B | $2.55 \pm 0.01$ | $2.57 \pm 0.11$ Z=−0.5; $p > 0.05$ | $2.35 \pm 0.01^{**}$ Z=−3.9; $p < 0.000$ | $2.37 \pm 0.01$ Z=−1.2; $p > 00.5$ |
| Phosphorus, mmol/l | A | $1.10 \pm 0.01$ | $1.23 \pm 0.01^{*}$ Z=−3.6; $p < 0.000$ | $0.94 \pm 0.03^{**}$ Z=−3.2; $p < 0.001$ | $0.97 \pm 0.02$ Z=−1.0; $p > 0.05$ |
| | B | $1.11 \pm 0.03$ | $1.17 \pm 0.03^{*}$ Z=−3.5; $p < 0.000$ | $1.14 \pm 0.03$ Z=−0.6; $p > 0.05$ | $0.99 \pm 0.01^{*}$ Z=−3.3; $p < 0.01$ |
| Cholesterol, mmol/l | A | $4.32 \pm 0.13$ | $4.72 \pm 0.08^{*}$ Z=−3.9; $p < 0.000$ | $4.10 \pm 0.12$ Z=−0.2; $p > 0.05$ | $4.15 \pm 0.09$ Z=−1.0; $p > 0.05$ |
| | B | $4.20 \pm 0.06$ | $4.07 \pm 0.04^{*}$ Z=−2.4; $p < 0.05$ | $4.37 \pm 0.06^{**}$ Z=−2.3; $p < 0.05$ | $4,32 \pm 0,04$ Z=−1.1; $p > 0.05$ |
| LDH, c. u. | A | $371.50 \pm 7.42$ | $502.55 \pm 7.44^{*}$ Z=−3.9; $p < 0.000$ | $434.55 \pm 16.84^{**}$ Z=−2.4; $p < 0.05$ | $447.75 \pm 7.98$ Z=−0.9; $p > 0.05$ |
| | B | $389.50 \pm 6.52$ | $442.20 \pm 18.26^{*}$ Z=−2.2; $p < 0.05$ | $378.50 \pm 15.97$ Z=−0.5; $p > 0.05$ | $379.25 \pm 9.25$ Z=−0.1; $p > 0.05$ |

**Notes.**
*The difference in comparison with previous results is significant according to Wilcoxon's test ($p < 0.05$).
**The difference between basal parameters (before training load) in comparison with previous results is significant according to Wilcoxon's test ($p < 0.05$).

We recorded increased concentration of cholesterol in the blood serum of group A athletes in response to physical stimuli at the beginning of the study by 9.2% ($p < 0.05$). On the contrary, group B athletes demonstrated decrease in cholesterol concentration at all stages of control by 3.9% ($p < 0.05$). After 2 months of training, group B had changes in the studied indicator, but the results were not reliable.

Table 5 provides the results of changes in the activity of LDH in the blood serum of MMA athletes. Using medium intensity training load regime ($R_a = 0.64$) required almost twice the energy expenditure in response to stress stimuli at the beginning of the study compared with the results found in conditions of using the regime of high intensity training load ($R_a = 0.72$). There were significant adaptive body changes in group B athletes, who used the regime of high intensity training load, on the background of lowering the basal level of LDH in their blood serum.

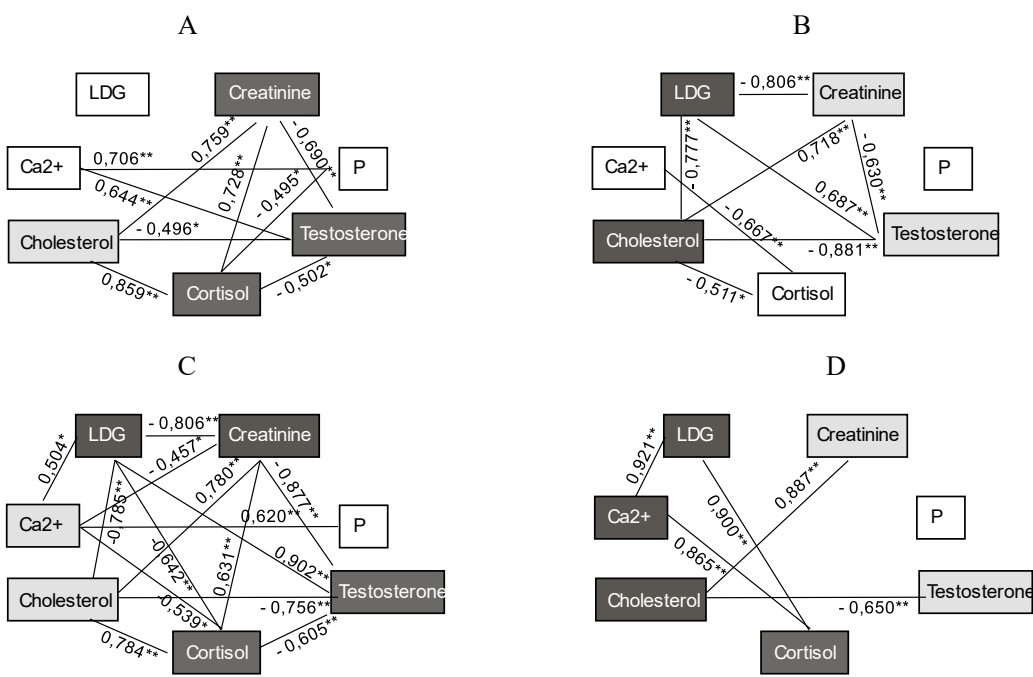

**Figure 1** **Correlation between biochemical parameters of blood in group A athletes during 2 months of research in the conditions of medium intensity training load regime ($Ra = 0.64$).** (A) The results recorded at the beginning of the study before training; (B) the results recorded at the beginning of the study after training; (C) the results recorded after 2 months of study before training; (D) the results recorded after 2 months of study after training; two asterisks (**) indicates a correlation at the level of 0.01; one asterisk (*) indicates a correlation at the level of 0.05.

Based on laboratory monitoring of blood biochemical indicators in athletes of both groups, samples were taken at rest and after training at the beginning and end of the study, we obtained the corresponding results of correlations using Spearman's rank correlation coefficient. The obtained data allow to determine which of the studied biochemical parameters of blood have the greatest number of strong correlations and can be used as a set of informative markers for assessing adaptive-compensatory reactions in MMA athletes of strike fighting style (Figs. 1–2).

The obtained results analysis showed that before long-term use of different intensity regimes of training load ($R_a = 0.64$ and $R_a = 0.72$) the informative biochemical markers for assessing the condition of athletes before training were cortisol, testosterone, and creatinine in both groups. Our study proved that at the beginning of the research the biochemical criteria for assessing adaptive-compensatory responses to physical stimuli were LDH and cholesterol in athletes of group A, and LDH, testosterone and cortisol in group B athletes.

The laboratory tests and correlation analysis after 2 months of training established that the set of biochemical markers indicating adaptation completely changed in athletes of group B and consisted of LDH, phosphorus, cholesterol, and calcium. Moreover, the set of biochemical criteria for assessing adaptive-compensatory reactions (after training)

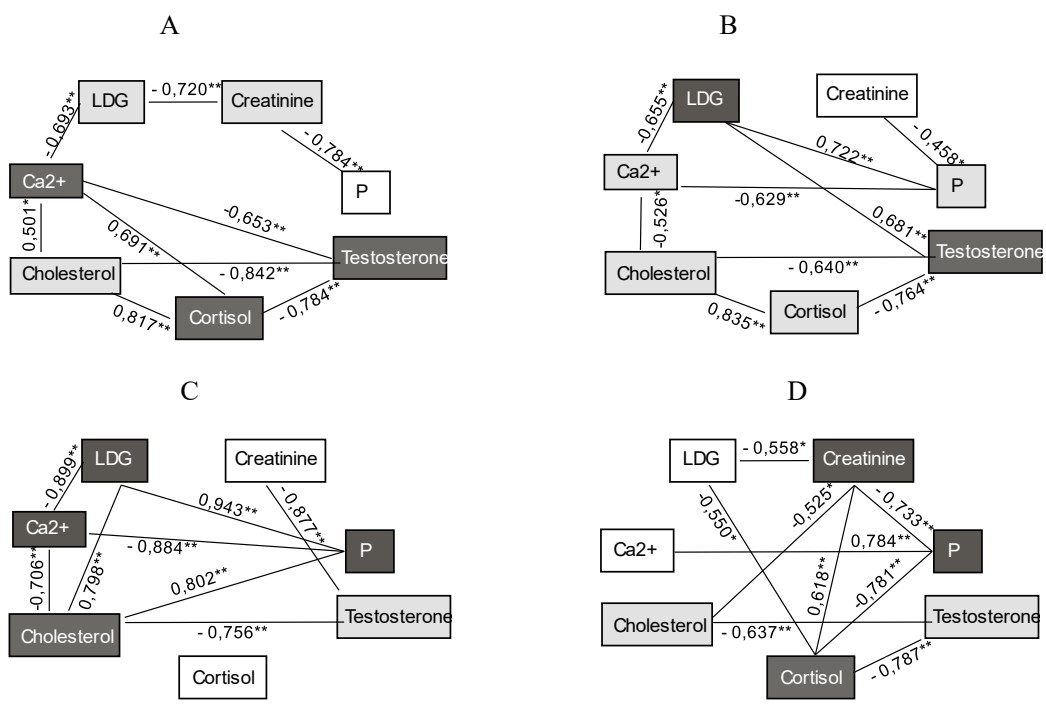

**Figure 2** (A–D) Correlation between biochemical parameters of blood in group B athletes during 2 months of research in the conditions of high intensity training load regime ($Ra = 0.72$).

also changed in group B athletes in response to high intensity training load ($R_a = 0.72$), compared with the data recorded at the beginning of the study.

Figures 1 and 2 show that the concentration of cortisol, testosterone, creatinine and LDH activity are informative biochemical markers of group A athletes recorded before training. The results fixed after training indicated that the biochemical parameters of LDH, cholesterol and calcium, cortisol in serum were indicators of assessing adaptive-compensatory reactions of group A participants.

## DISCUSSION

The lack of a clear understanding of the overall strategy for optimizing the comprehensive control system in mixed martial arts is one of the controversial issues for scientists in physical education and sports physiology (*James et al., 2016a*; *James et al., 2016b*; *Kirk et al., 2021*; *Chernozub et al., 2019*; *Tota & Wiecha, 2022*). One of the directions of this problem is related to the need to identify the most informative criteria for assessing the adaptive body changes in MMA athletes of strike fighting style and to find effective regimes of power training in the process of special training. Despite the generally accepted system of performance control in world practice, mainly by assessing the dynamics of morphometric parameters, changes in maximum muscle strength parameters (1RM) and special training (*Del Vecchio & Franchini, 2011*; *Schoenfeld et al., 2016*; *James et al., 2016a*; *James et al., 2016b*; *Loturco et al., 2021*), in recent years scientists (*Marques et al., 2017*;

*Walker et al., 2017*; *Crewther, Obmiński & Cook, 2018*; *Stajer, Vranes & Ostojic, 2020*; *Sarin et al., 2019*) have paid attention to the use of biochemical blood markers to determine the course of adaptive changes.

The bioimpedansometry results proved that using the most effective for power fitness regime (Ra = 0.72) contributed to a significant reduction in body fat on the background of increasing fat-free mass. Changes in these indicators of body composition point at a pronounced process of adaptation of the organism (*e.g.*, *Chernozub et al., 2019*).

On the background of increasing maximum muscle strength and reducing the level of body fat in group B athletes, the parameters of special training increased 4 times compared with group A results. We assume that these changes are connected with increased adaptation reserves due to intra-muscular and intermuscular coordination and energy efficiency of muscular activity in anaerobic exercise regime (*James et al., 2016a*; *James et al., 2016b*; *Futorniy et al., 2016*).

The peculiarities of changes in the steroid hormones cortisol and testosterone in the serum of MMA athletes with a strike fighting style in response to training loads of different intensity, confirmed the research results in boxing and weightlifting where eccentric exercises were used during training (*Philippou et al., 2017*; *Crewther, Obmiński & Cook, 2018*; *Kılıç et al., 2019*). However, the results of our research contradict the research results in athletics (*Wahl et al., 2013*); which stated that prolonged use of high intensity training load regime helped to reduce basal testosterone level in the blood serum. In addition, the increase in basal testosterone and LDH in the blood serum after training load of medium and low intensity in conditions of predominantly aerobic energy supply, indicates the activation of compensatory mechanisms in athletes (*e.g.*, *Shin et al., 2016*; *Wochyński & Sobiech, 2017*). That is why the use of basal testosterone as a marker for assessing adaptive changes during high intensity training load regime requires further research.

We discovered that the nature of changes in creatinine in the blood serum of both group participants before and after training load of different intensity during 2 months of research, confirmed the results of experimental studies in power fitness (*Titova et al., 2018*; *Chernozub et al., 2020*). At the same time, the increase in basal creatinine level, which indicates growing resources for creatine phosphokinase response on the background of increasing fat-free mass and decreasing body fat, confirm the research results of a number of scientists (*Clarkson et al., 2006*; *Papassotiriou & Nifli, 2018*; *Stajer, Vranes & Ostojic, 2020*). These researchers recorded an increase in the concentration of creatinine in the serum during anaerobic lactate regime of energy supply of muscular activity in athletics and wrestling training in the process of eccentric exercises of high intensity.

Lowering blood cholesterol levels in athletes of group B in response to high intensity training load (Ra = 0.72) confirm the results of studies concerning strength training (*Costa et al., 2019*; *Sarin et al., 2019*). However, these studies were not focused on the parameters of the volume and intensity of strength training, but only assessed the overall impact on training with weights on certain biochemical markers of blood. The concentration of cholesterol in the blood of group A athletes increased, which may be due to compensatory reactions of the body in response to this stress stimulus.

Increases in blood calcium were observed only in response to high intensity training load, regardless the stage of the study. However, our data differ from the results of research in power sports (*e.g.*, *Pullinger et al., 2019*), which show a decrease in the concentration of calcium ions in response to medium and low intensity exercise. Such changes in this biochemical indicator slow down the transmission of nerve impulses, which negatively affects the speed of muscle contraction.

Investigating the structure of complex control in martial arts, a number of researchers (*Chernozub et al., 2019*; *Kirk et al., 2021*; *Tota & Wiecha, 2022*) found out that changes in biochemical indicators of blood were the most informative markers of adaptation compensatory reactions of athletes to stressful physical stimuli. Taking into account the complexity and high cost of conducting comprehensive biochemical blood tests to study adaptation processes in different training load regimes, most leading experts in biology and sports (*Marques et al., 2017*; *Philippou et al., 2017*; *Wochyński & Sobiech, 2017*) used only a small number of markers (2–4). However, even with a clear record of changes in the activity of certain physiological processes in response to stress stimuli, it is almost impossible to establish the nature of adaptive or compensatory reactions using a standard set of biochemical blood markers in such studies.

In this study we obtained the results and the peculiarities of changes in the biochemical blood parameters during special training in MMA and determined the correlation between them, which allowed to establish a minimum set of criteria for assessing adaptation to different training load regimes. A similar direction of research concerning adaptive-compensatory body reactions to power loads of different intensity was conducted with the intention to optimize the training system of MMA athletes and in other sports (*James et al., 2016a*; *James et al., 2016b*; *Crewther, Obmiński & Cook, 2018*; *Chernozub et al., 2019*; *Chernozub et al., 2020*).

## CONCLUSIONS

Using high intensity training load regime by MMA athletes with strike fighting style in the process of special strength training allows to increase the parameters of muscular strength and the number of accurate kicks in the shortest possible time. The implementation of this study results in competitive activity of MMA athletes will increase the effectiveness of attacking and counterattacking actions of athletes by increasing the power, speed and number of accurate kicks.

A wide range of biochemical blood indicators, which are used as criteria for assessing adaptive-compensatory reactions in the conditions of special strength training of MMA athletes, shows different tendencies to change depending on the intensity of the load. The results of laboratory tests and correlation analysis allows to clearly determine the optimal set of biochemical blood markers to assess the processes of short-term and long-term adaptation of athletes using medium and high intensity training loads.

A clear definition of the optimal set of criteria for assessing the adaptive-compensatory changes in MMA athletes of strike fighting style in the process of special strength training will allow quickly adjust the parameters of the load mode to accelerate the body's functionality.

## LIMITATIONS AND FUTURE STUDIES

The number of participants in the research groups was limited by the peculiarity of the training stage and the selection of the subjects. In the future it will be necessary to involve in research more athletes of different levels of training with the aim of using the obtained results in the process of long-term training. In further studies it is recommended to control samples every 20–30 days regardless of the possible long-term (up to 6–9 months) examination period for obtaining detailed determination of the course of adaptive and compensatory reactions to a physical stimulus at the expense of biochemical blood markers. Using indicators of athletes' heart rate variability in conditions of intense muscle activity will allow to improve the system of control over training loads and to upgrade the mechanisms of the training system correction.

## ACKNOWLEDGEMENTS

The article is a part of the planned scientific work "Development and implementation of innovative technologies and correction of the functional state of a person during physical activity in sports and physical therapy", (state registration number 0117U007145) and Ministry of Education and Science of Ukraine project number 0118U000809.

### Funding

The authors received no funding for this work.

### Competing Interests

The authors declare there are no competing interests.

### Author Contributions

- Andrii Chernozub conceived and designed the experiments, performed the experiments, analyzed the data, prepared figures and/or tables, authored or reviewed drafts of the article, and approved the final draft.
- Veaceslav Manolachi conceived and designed the experiments, performed the experiments, analyzed the data, authored or reviewed drafts of the article, and approved the final draft.
- Georgiy Korobeynikov conceived and designed the experiments, performed the experiments, analyzed the data, prepared figures and/or tables, authored or reviewed drafts of the article, and approved the final draft.
- Vladimir Potop conceived and designed the experiments, performed the experiments, analyzed the data, prepared figures and/or tables, authored or reviewed drafts of the article, and approved the final draft.
- Liudmyla Sherstiuk performed the experiments, analyzed the data, authored or reviewed drafts of the article, and approved the final draft.

- Victor Manolachi conceived and designed the experiments, performed the experiments, analyzed the data, authored or reviewed drafts of the article, and approved the final draft.
- Ion Mihaila conceived and designed the experiments, analyzed the data, prepared figures and/or tables, and approved the final draft.

## Human Ethics

The following information was supplied relating to ethical approvals (*i.e.*, approving body and any reference numbers):

The experimental study was approved by Petro Mohyla Black Sea National University Ethics Committee for Biomedical Research (Ethical Application Ref: ecbr10-02-2022).

## Data Availability

The raw measurements are available in the Supplementary Files.

## Supplemental Information

Supplemental information for this article can be found online at http://dx.doi.org/10.7717/peerj.13827#supplemental-information.

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
