# Peer review of "Criteria for assessing the adaptive changes in mixed martial arts (MMA) athletes of strike fighting style in different training load regimes"

_PeerJ, doi:10.7717/peerj.13827_

## Round 0.1 · original submission · Major Revisions

Thank you for submitting the manuscript to PeerJ. It has been reviewed by experts in the field and we request that you make major revisions before it is processed further.

We look forward to hearing from you soon.

Best wishes,

Badicu Georgian, Ph.D

·

Basic reporting

- The English language is at a satisfactory level throughout the manuscript.

- The criterion is partially fulfilled.

- The article lacks simplicity and clarity. On the other hand, readers will have no trouble understanding tables and figure.

- The obtained results are in accordance with the research hypothesis.

Experimental design

- The presented study is in line with the aims and scope of the journal.

- The criterion is fulfilled. The findings of this research are innovative and represent a significant progress in the field of training load design in combat sports athletes.

- The criterion is partially fulfilled.

- The methods section provided enough information for readers.

Validity of the findings

- The criterion is fulfilled.

- Fundamental data are unambiguous and are clearly presented to the readership.

- The conclusion is the weakest part of the entire study. The authors must once again check this section.

Additional comments

• The abstract is really extensive and contains too many words. The results within the mentioned section have several unnecessary claims.
• Lines 75-78: Can you provide references for this statement, please? Are there researchers who have previously applied listed indicators to assess the effectiveness of different training loads in athletes?
• The term "various/other sports" is frequent used in manuscript. It would be clearer for the readers if you would report more precisely which sports do you mean? Combat sports or?
• Lines 113-115 and 145-148: Do you mean these assertions are indispensable? Please, remove them from the text.
• Authors should change term "Methods" to "Measurements". In order to improve the readability and clarity of the text, it is also recommended to add subtitles for each separate measurements (maximal muscle strength, special training, body composition and biochemical parameters).
• Why did you explain the findings in the results section? This is more appropriate for discussion. Check all paragraphs again, please!
• Line 231: How can researchers be sure that the results obtained are occur due to the increased levels of intramuscular and intermuscular coordination of athletes? Are there any previous findings/references that can confirm this hypothesis? Additionally, this statement is also more appropriate for discussion.
• Line 240: The abbreviations in Table 4 should be placed below it.
• Please improve the visibility of Figures 1 and 2. Check the text above the figures again, please.
• Line 287: "The obtained results" – The mentioned terms are often used in the results section, which will be boring for the readership.
• Line 298: Please report results for group A that implemented a medium intensity training load.
• The discussion requires certain corrections. The authors analyzed only biochemical parameters, and what about other measured indicators?
• Lines 330-332: The mentioned contradictory findings need to be explained.
• Lines 360-364: This statement should be moved to discussion.
• Lines 365-380: You just repeated the results obtained. Please, remove this text from the conclusion section.

Reviewer 2 ·

Basic reporting

The evaluated paper investigates the adaptive reactions of young athletes (MMA fighters) to 2 regimes of effort (medium and high intensity and load). The value of the study results from the high number of measured and analyzed parameters (data related to the accuracy of the movements execution technique, body composition, biochemical indicators of blood and analysis of the associations between these independent, studied variables). Laboratory investigations are the strong point of this study. The research is useful for the scientific direction of the training process for the targeted athlete category.

The reviewed work is valuable and useful to those working in the field of sports (especially due to detailed laboratory explorations) and may be published after a few minor revisions.

Experimental design

I think it would be useful to formulate working hypotheses (at the end of the introduction), to which the results from the tables and graphs should be reported.

Please specify the (MMA) sports activity experience of the 2 groups studied and some data related to the height and weight of these athletes (average and standard deviation).

Could you indicate the 2-month interval in which you performed the experiment (months and year)?

In what stage of training (general physical training, pre-competitive or competitive training) were the investigated fighters?

Please specify the type of study performed (longitudinal or cross-sectional study).

Validity of the findings

Why didn't you compare the results between the 2 groups (A and B as independent samples) by the non-parametric Mann-Whitney U test? I noticed that you have a large volume of analyzed data anyway (and these additional comparisons would have doubled the number of tables), but it would be advisable to make these comparisons perhaps in another paper, for a complete view of your results. However, this does not reduce the value of the study.

Tables 2-5 show the results of the Friedman-Anova test and compare the differences between the 3 tests (T1 - initial, T2 - intermediate - one month and T3 - final - 2 months). It should result in 3 data pairs with 3 values of Z and the associated significance thresholds: T1-T2, T1-T3 and T2-T3. Only two pairs appear in all tables, probably T1-T2 and T2-T3. Could you include the missing pair?

You used SPSS software, so the effect size value (expressed by Kendall W) might have been calculated if this option was checked. It would provide additional support in arguing for the significance of the differences obtained. It's just a remark, the study is well-argued enough even without this statistical indicator.

In Table 4, for the BF indicator (body fat %) there are big differences between groups A and B, right from the initial testing. What is the explanation?

Figure1 and 2- The title: Correlation between biochemical parameters of blood in group A athletes before (A and C) and after training load (B and D) during 2 months of research in the conditions of medium intensity training load regime (Ra = 0.64). Below figures, there appear in the initial tests A and B, respectively in the final ones (after 2 months) C and D.

Additional comments

You refer to the morpho-functional parameters (line 22). This wording / expression is repeated in "Participants" (line 114), respectively in "Conclusions" (line 359). Anthropometric parameters (height, body weight, perimeters and diameters etc. are not presented in this study).

You could include a short paragraph to identify issues related to the specificity of physical effort in MMA (which differentiates it from other combat sports).

Line 245: Table 5 presents the results of changes in biochemical inadicators of the blood serum…. I think the word indicators is correct.

---

## Round 0.2 · Major Revisions

Please send a cover letter with each reviewer comment and response. Do not submit comments and replies separately.

---

## Round 0.3 · Minor Revisions

Before publishing, please include study limitations and future research

·

Basic reporting

The authors fulfilled all criteria relating to the "Basic reporting" section.

Experimental design

The authors successfully responded to comments associated with the "Experimental design" section.

Validity of the findings

The authors fulfilled all criteria relating to the "Validity of the findings" section.

Additional comments

I have no additional comments.

Reviewer 2 ·

Basic reporting

I believe that the explanations mentioned by the authors and the changes made in this scientific article would allow publication in this version.

Experimental design

The authors made most of the suggested changes, or provided reasoned explanations, when they had a different approach to the issues previously reported.

Validity of the findings

The authors made most of the suggested changes, or provided reasoned explanations, when they had a different approach to the issues previously reported.

---

## Round 0.4 · accepted · Accept

The manuscript is ready to be accepted.